# Separation of Five Iridoid Glycosides from *Lonicerae Japonicae* Flos Using High-Speed Counter-Current Chromatography and Their Anti-Inflammatory and Antibacterial Activities

**DOI:** 10.3390/molecules24010197

**Published:** 2019-01-07

**Authors:** Ran Yang, Lei Fang, Jia Li, Zhenhua Zhao, Hua Zhang, Yongqing Zhang

**Affiliations:** 1Key Laboratory of Natural Pharmaceutical Chemistry, Shandong University of Traditional Chinese Medicine, Jinan 250200, China; cpu1045405@126.com (R.Y.); fleiv@163.com (L.F.); LJYTL7172@163.com (J.L.); szu@126.com (Z.Z.); 2School of Biological Science and Technology, University of Jinan, Jinan 250022, China

**Keywords:** *Lonicerae Japonicae* Flos, high-speed counter-current chromatography, iridoid glucosides, anti-inflammatory activity, antibacterial activity

## Abstract

A high-speed counter-current chromatography (HSCCC) method, using a two-phase solvent system composed of ethyl acetate/*n*-butanol/methanol/water (5:1:1:5, *v*/*v*/*v*/*v*), was successfully established to separate the five iridoid glucosides 7-*O*-ethyl sweroside (**1**), secologanin dimethylacetal (**2**), adinoside F (**3**), (*7R*)-secologain *n*-butyl methyl acetal (**4**) and adinoside G (**5**) from *Lonicerae Japonicae* Flos. Their purities were 96.8%, 98.5%, 93.3%, 98.0% and 99.9%, respectively. All the iridoid glucosides were identified by HR-ESI-MS, 1D and 2D NMR. Compounds **3** and **5** are new iridoid glucosides. The anti-inflammatory tests showed that compounds **1**–**5** all expressed moderate inhibitory effects on *β*-glucuronidase release in rat polymorphonuclear leukocytes (PMNs) induced by platelet-activating factor (PAF) with IC_50_ values ranging from 4.52 to 6.50 µM, while the antibacterial assays demonstrated that all the compounds displayed mild inhibitory activities against *Staphylococcus aureus ATCC 25923* with MIC values ranging from 13.7 to 26.0 µg/mL.

## 1. Introduction

*Lonicerae Japonicae* Flos, known as Jinyinhua in the Chinese Pharmacopeia, refers to the dried flower buds or the initial flowers of *Lonicera japonica* Thunb. It has been widely used in traditional Chinese medicine to treat various diseases, such as cough, thirst, arthritis, fever, sore throat, and influenza infection [1]. In addition, it can also been used in functional foods, cosmetics and other applications [2]. Modern phytochemistry and pharmacology studies have shown that chlorogenic acids, iridoid glucosides, polyphenols, flavonoids, triterpenoid saponins, organic acids and volatile oils were the active ingredients in *Lonicera* Species [3,4,5]. Iridoid glucosides, as a special class of monoterpenoids, have been frequently reported to display a variety of pharmacological properties such as anti-inflammatory [6], antibacterial [7], anti-virus [8], anti-allergic [9], and hepatoprotective effects [10], and play important roles in the known effects of *L. Japonicae* Flos. 

The conventional methods to isolate iridoid glucosides from *L. Japonicae* Flos include silica gel chromatography, polyamide column chromatography and others, but most of these methods lead to low yields and unideal recoveries due to the repeated chromatographic steps and the adsorption of samples on the solid supports [11]. Considering these problems, it is necessary to develop a fast and more efficient method to isolate and separate iridoid glucosides from *L. Japonicae* Flos. High-speed counter-current chromatography (HSCCC) was first developed in the early 1980s [12]. It is a kind of chromatographic technology based on the theory of liquid-liquid distribution, and has been widely used in the isolation of natural products for its advantages of high recovery, simple sample preparation, high efficiency as well as ideal repeatability [13]. However, as far as we know, there have been no reports on the isolation and separation of iridoid glucosides by HSCCC from *L. Japonicae* Flos. In our present study, five iridoid glucosides (Figure 1), 7-*O*-ethyl sweroside (**1**), secologanin dimethylacetal (**2**), adinoside F (**3**), (*7R*)-secologain *n*-butyl methyl acetal (**4**) and adinoside G (**5**), were successfully isolated and separated from *L. Japonicae* Flos by a HSCCC method, and compounds **3** and **5** were identified as new iridoid glucosides.

## 2. Results and Discussion

### 2.1. Selection of Two-Phase Solvent Systems

In the HSCCC separation, it is particularly important to choose a solvent system with suitable *K* values (0.5 < *K* < 2.0) for target compounds [14]. In the present study, two groups of solvent systems (ethyl acetate/*n*-butanol/water and ethyl acetate/*n*-butanol/methanol/water) with different proportions were tested. The results are listed in Table 1. 

It proved hard to isolate and separate target compounds when solvent systems composed of ethyl acetate/*n*-butanol/water (2:1:3, 4:1:5 and 11:1:12, *v*/*v*/*v*) were used, as the *K* values were small, especially for compounds **1**, **2** and **3**. Then, systems composed of ethyl acetate/*n*-butanol/methanol/water (from 11:0.5:0:5:11 to 5:1:1:5, *v*/*v*/*v*/*v*) were tested. The *K* values of compounds **1** and **2** were less than 0.5 with the ratios of 11:0.5:0.5:11 and 11:0.5:1:11 (*v*/*v*/*v*/*v*), resulting in a poor partition of target compounds in the mobile phase. The *K* value of compound **5** was too large with the ratio of 11:3:0.5:11 (*v*/*v*/*v*/*v*), which might lead to a long elution time. When the ratio was adjusted to 5:1:1:5 (*v*/*v*/*v*/*v*), the *K* values of all the target compounds fell into a suitable range. In addition, the retention of the stationary phase was 48.5% in this system, indicating that there was sufficient stationary phase to allow the sample to be distributed. The HSCCC chromatogram was listed in Figure 2, where compounds **1** (peak I, collected during 127–136 min, 23 mg), **2** (peak II, collected during 148–159 min, 28 mg), **3** (peak III, collected during 287–311 min, 15 mg), **4** (peak IV, collected during 358–393 min, 9 mg) and **5** (peak V, collected during 497–550 min, 27 mg) were obtained from 150 mg of crude sample in one step within 11 h. However, peaks 200 and 250 min contained more than one compound by HPLC analysis. 

The HPLC chromatograms of the crude sample and all the target compounds were shown in Figure 3. The purities of compounds **1**–**5** were recorded as 96.8%, 98.5%, 93.3%, 98.0% and 99.9%, respectively, according to the HPLC peak-area percentages. The results of this study thus showed that HSCCC can successfully isolate and separate compounds **1**–**5** from *L. Japonicae* Flos.

### 2.2. Identification of Compounds

Two new iridoid glucosides, adinoside F (**3**) and adinoside G (**5**), together with three known ones (**1**, **2**, and **4**), were determined by the HR-ESI-MS, 1D and 2D NMR data (see Appendix A). 

Adinoside F (**3**) was separated as a yellow crystalline solid with [*α*]^25^_D_ − 109.3 (CH_3_OH, *c*
*=* 1.0). Its molecular formula was deduced by HR-ESI-MS at *m/z* 481.1680 [M + Na]^+^ (calculated for C_21_H_30_NaO_11_, 481.1686). Its IR spectrum displayed absorption bands of hydroxyl (3441 cm^−1^) and carbonyl (1712 cm^−1^) groups. Its ^1^H and ^13^C-NMR spectra (Table 2 and Table 3) were closely similar to those of adinoside A [15], except for the presence of an ethoxy group in **3** [*δ*_H_ 1.24 (t, *J* = 7.4 Hz, H-4′′) and 4.11 (m, H-3′′); *δ*_C_ 14.5 (C-4′′) and 61.8 (C-3′′)]. The location of the ethoxy group was determined by the correlations from H-3′′ to C-1′′ in the HMBC spectrum (Figure 4). The correlations of H-1/H-6, H-8 and H-5/H-9 in the NOESY spectrum (Figure 5) suggested H-5 and H-9 were on the same side, while H-1 on the other side. Therefore, compound **3** had the same relative configurations with secologanin [16] and can be named as adinoside F.

Adinoside G (**5**) was obtained as yellow powder with [*α*]^25^_D_ − 56.1 (CH_3_OH, *c*
*=* 0.6). Its molecular formula was indicated from HR-ESI-MS at *m/z* 509.1990 [M + Na]^+^ (calculated for C_22_H_34_NaO_11_, 509.1999). Its IR absorption bands showed the existence of hydroxyl (3428 cm^−1^) and carbonyl (1712 cm^−1^) moieties. Its ^1^H and ^13^C-NMR spectral features (Table 2 and Table 3) of **5** were very similar to compound **3**. The difference between them was the replacement of the ethoxy group in **3** by an *n*-butyl group in **5**, as supported by the HMBC correlations from H-3′′ to C-1′′ and from H-6′′ to C-4′′ (Figure 4). Furthermore, the NOESY correlations in suggested that its relative configurations were the same with compound **3**. Thus, compound **5** was named as adinoside G.

The three known ones were identified as 7-*O*-ethyl sweroside (**1**) [17], secologanin dimethylacetal (**2**) [18] and (*7R*)-secologain *n*-butyl methyl acetal (**4**) [19] by comparing their spectroscopic data with the literature.

### 2.3. Anti-Inflammatory and Antibacterial Activities

All the compounds were evaluated for anti-inflammatory and antibacterial activities. Compounds **1**–**5** all exhibited moderate anti-inflammatory activities to suppress the release of *β*-glucuronidase with IC_50_ values ranging from 4.52 to 6.50 µM (Table 4), while they also displayed mild antibacterial activities against *Staphylococcus aureus ATCC* 25923 with MIC values ranging from 13.7 to 26.0 µg/mL (Table 5).

## 3. Materials and Methods

### 3.1. Plant Materials and Reagents

Fresh *L. Japonicae* Flos samples were collected at Jinan, Shandong Province, China, in 2017, and identified by Professor Jia Li (Shandong University of Traditional Chinese Medicine). A voucher specimen (No JYH-201705) was stored in the Key Laboratory of Pharmaceutical Chemistry, University of Jinan, China. Methanol (HPLC grade) was obtained from the Oceanpak company (Goteborg, Sweden). Distilled water was prepared using a Milli-Q system (Billerica, MA, USA). Other reagents including ethyl acetate, dichloromethane, *n*-butanol, ethanol and methanol of analytical grade were purchased from Fuyu Chemical Factory (Tianjin, China).

### 3.2. Instruments

The TBE-300C HSCCC instrument (Shanghai Tauto Biotech Co., Ltd., Shanghai, China) used in our present study was equipped with a sample loop (volume: 20 mL) and a three preparative polytetrafluoroethylene coils (volume: 300 mL, radius: 1.3 mm). The *β* value of the multilayer coil ranged from 0.5 to 0.8 at the internal and external, respectively (*β*= r/R, where R is the distance of the centrifuge from holder axis to central axis, and r is the spacing between the coil and holder shaft). The revolution speed can be regulated from 0 to 900 rpm. In addition, a TBP-5002S constant flow pump, a TBD2000 UV monitor, a DC-0506 constant low-temperature bath as well as a HW-2000 workstation were all provided by Tauto Biotech Co., Ltd. (Shanghai, China).

An Agilent 1260 system (Agilent Technologies, Santa Clara, USA) was used in the HPLC analysis. It was equipped with a G1316A column thermostat, a G1314B UV-vis photodiode array detector) and a G1313A auto-sampler. An Agilent Technologies 6250 Q-TOF LC/MS was used to detect HR-ESI-MS data. A Rudolph VI digital polarimeter (Rudolph Research Analytical, Hackettstown, USA) was employed to record optical rotation data. In addition, an AVANCE DRX (600 MHz) (Bruker, Billerica, MA, USA) nuclear magnetic resonance (NMR) spectrometer was used to obtain 1D and 2D NMR spectra.

### 3.3. Preparation of Crude Sample

Firstly, the fresh flowers were treated with a vacuum freeze-drier, and then the freeze-dried materials (6 kg) were extracted with 95% ethanol (30 L) at room temperature. After being concentrated in vacuo, the ethanol extract (0.9 kg) was suspended in water and extracted with ethyl acetate (10 L) and *n*-butanol (12 L). The *n*-butanol soluble fraction (64.5 g) was first separated over a macroporous adsorbent resin column. The column was eluted with water-ethanol (100:0, 70:30, 50:50, 30:70, 10:90, *v*/*v*), yielding five fractions. Then the 50% ethanol eluate (7.5 g) was ready to be separated by HSCCC.

### 3.4. Selection of Two-Phase Solvent Systems and Preparation of the Sample Solution

The selection of solvent system is based on the partition coefficient (*K*) value, which can be measured by HPLC as follows. A bit of sample was dissolved with equal amounts of the upper and lower phases of the solvent system, and then, the same amounts of upper and lower solutions were taken for HPLC analysis, respectively. The peak areas of target compound in upper and lower phases were recorded as A_upper_ and A_lower_, respectively, and the *K* value was calculated as A_upper_/A_lower_. In this work, the optimal solvent system composed of ethyl acetate/*n*-butanol/methanol/water (5:1:1:5, *v*/*v*/*v*/*v*) was prepared to obtain the upper and lower phases. Then, 150 mg of dried sample was dissolved with solvent mixture which containing 7.5 mL of upper phase and the same volume of lower phase.

### 3.5. Procedure of HSCCC Separation

Firstly, the stationary phase (upper phase) was pumped into the multilayer coil with a flow rate of 50 mL/min, and then the operating speed was adjusted at 850 rpm. Next, the mobile phase (lower phase) was pumped at a flow rate of 2.0 mL/min. When the hydrodynamic equilibrium was reached, the sample solution was injected into the apparatus. The peak fractions were collected according to elution profiles at 230 nm. In the end of the process, the retention of the stationary phase was recorded as A_1_/A_2_, where A_1_ was the volume of the stationary phase pumped out from the column, and A_2_ was the total column volume.

### 3.6. Analysis and Identification of Peak Fractions 

The HPLC analysis of the crude sample and each peak fraction were performed as follows: an YMC-Pack ODS-A column (5 μm, 100 × 4.6 mm) was used with gradient elution at a flow rate of 1.0 mL/min. The mobile phase was methanol (A) and water (B) (0–0 min, 10–80% A). The work of identification was carried out by HR-ESI-MS and NMR data. 

### 3.7. Anti-Inflammatory Assay

Anti-inflammatory activities were evaluated as the inhibition of *β*-glucuronidase release caused by platelet activating factor (PAF) in rat polymorphonuclear leukocytes (PMNs). The test sample was prepared with a concentration of 0.1 M, and then diluted to 1 mM. Then the test sample (5 µL) and rat PMNs suspension (2.5 × 10^6^ cell/mL, 245 µL) were added into a test tube, and the tube was incubated at 37 °C for 20 min. Subsequently, 1 mM cytochalasin B (2.5 µL) was added into the tube with incubation for 10 min, and then 0.2 mM PAF (2.5 µL) was added for another 10 min. The reaction was terminated by transferring the tube to an ice bath. The mixture was centrifuged at 4000 rpm for 5 min, and the supernatant was obtained [20]. 

The supernatant (25 µL) and 2.5 mM phenolphthalein glucuronic acid (25 µL) were added into a 96-well plate, and 0.1 M HOAc (100 µL) was used as buffer in each well. The plate was incubated in 37 °C for 18 h. The reaction was stopped by adding 0.3 M NaOH solution (150 µL). The results were read at 550 nm, and the absorption values of PAF, test samples and control samples were recorded as A_PAF_, A_t_, and A_c_, respectively. Inhibition rate (IR) was calculated as IR (%) = (A_PAF_ − A_t_) − (A_PAF_ − A_c_) × 100%. The IC_50_ value was defined as the half maximal inhibitory concentration and Ginkgolide B was used as the reference. 

### 3.8. Antibacterial Assay

Antibacterial activities were tested with gram-positive strains (*Staphylococcus aureus ATCC 25923*, *Mycobacterium smegmatis ATCC 607*) and gram-negative strains (*Pseudomonas aeruginosa ATCC 27853*, *Escherichia coli ATCC 25922*). Penicillin was used as the positive control group. Firstly, all the strains were inoculated on LB medium with 37 °C for 24 h. Then the cell density was diluted to 1 × 10^5^ cfu/mL. The sample solution was added into the bacterial suspension in 96-well plate and the mixture was incubated for 24 h at 37 °C. Each sample solution was tested in dilution series ranging from 100 to 0.20 µg/mL. The optical density was measured at 600 nm with a Tecan Spark microplate reader (Tecan Trading Co., Ltd., Shanghai, China). The minimum inhibitory concentration (MIC) values for the tested strains were recorded [21]. 

## 4. Conclusions

In conclusion, a HSCCC method, using a two-phase solvent system composed of ethyl acetate/*n*-butanol/methanol/water (5:1:1:5, *v*/*v*/*v*/*v*), was successfully established to isolate and separate iridoid glucosides from *L. Japonicae* Flos. Five iridoid glucosides, containing two new ones (**3**: adinoside F and **5**: adinoside G), were obtained in one step, and their purities were 96.8%, 98.5%, 93.3%, 98.0% and 99.9%, respectively. All the five iridoid glucosides showed moderate anti-inflammatory effects and mild inhibitory activities against *Staphylococcus aureus ATCC 25923*. This study provided a faster and more efficient method to isolate and separate bioactive iridoid glucosides from *L. Japonicae* Flos, and that will be helpful in our further research.

## Figures and Tables

**Figure 1 molecules-24-00197-f001:**
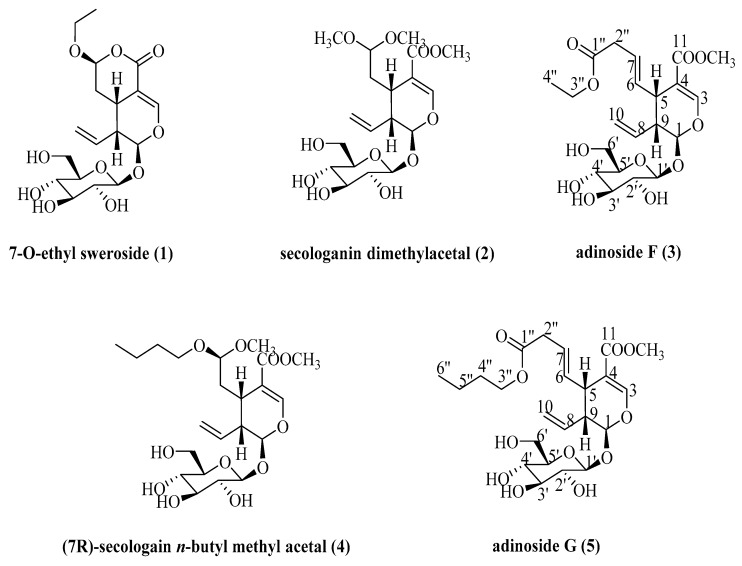
Chemical structures of compounds **1**–**5**.

**Figure 2 molecules-24-00197-f002:**
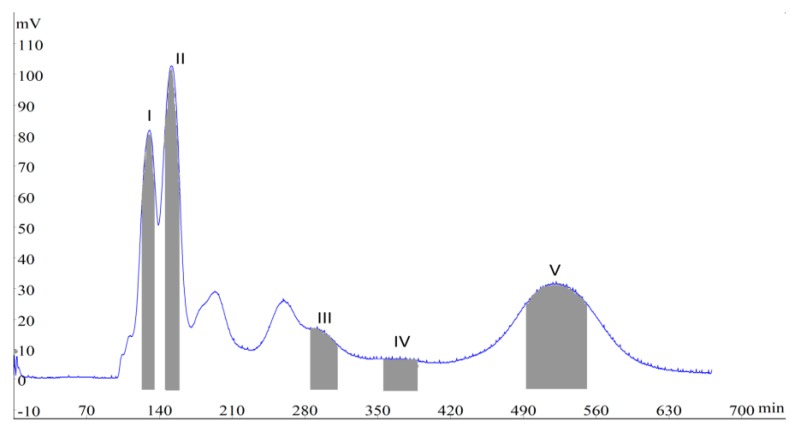
HSCCC chromatogram of the crude sample from *L. Japonicae* Flos. Two-phase solvent system: ethyl acetate/*n*-butanol/methanol/water (5:1:1:5, *v*/*v*/*v*/*v*); stationary phase: upper phase; mobile phase: lower phase; flow-rate: 2.0 mL/min; revolution speed: 850 rpm; detection wavelength: 230 nm; separation temperature: 25 °C.

**Figure 3 molecules-24-00197-f003:**
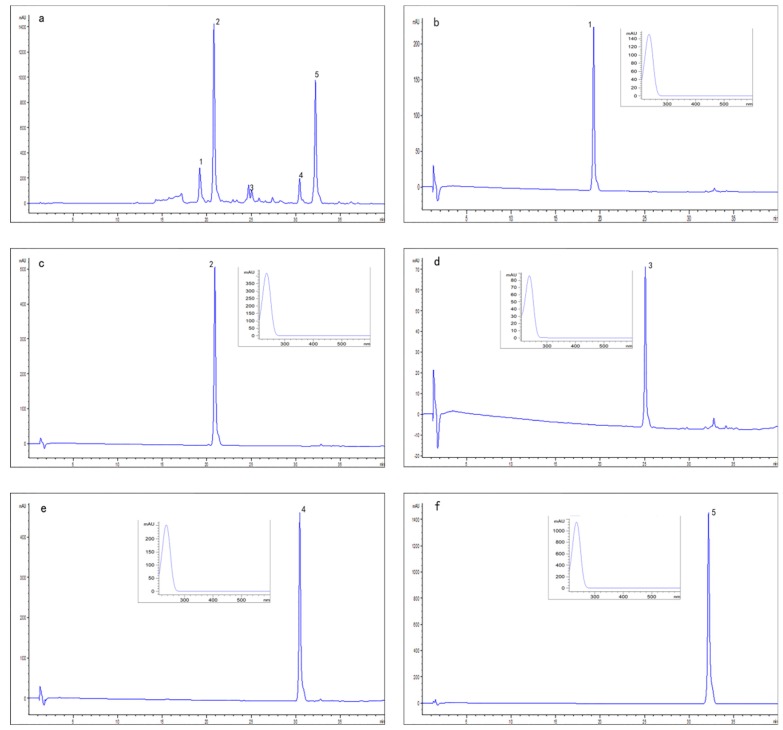
(**a**) HPLC analysis of the crude sample; (**b**–**f**) HPLC chromatograms and UV spectra of compounds **1**–**5**. Column: YMC-Pack ODS-A column (5 μm, 100 × 4.6 mm); mobile phase: methanol (A)-water (B) (0–40 min, 10–80% A); flow rate: 1.0 mL/min; detection wavelength: 230 nm; column temperature: 25 °C.

**Figure 4 molecules-24-00197-f004:**
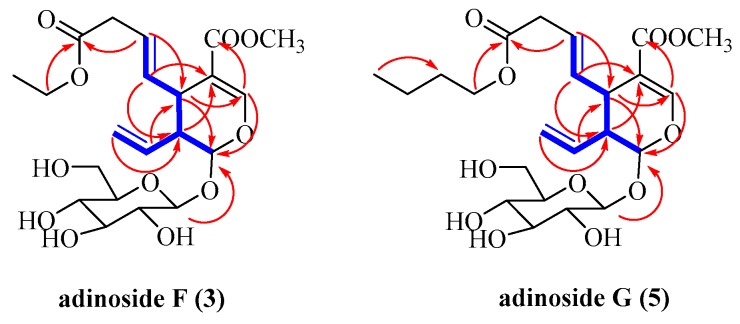
^1^H-^1^H COSY (blue bold bonds) and key HMBC correlations (red arrows) of compounds **3** and **5**.

**Figure 5 molecules-24-00197-f005:**
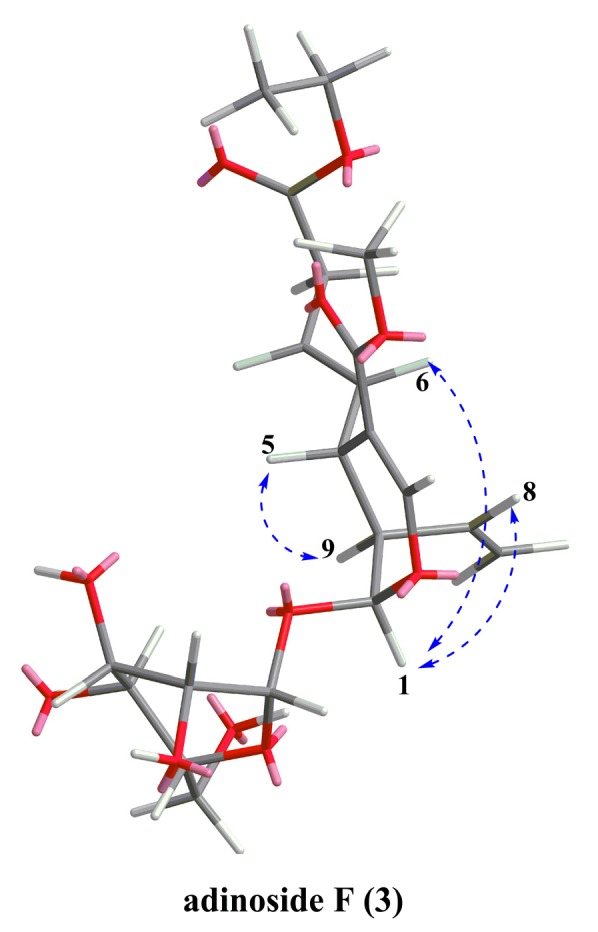
Key NOEs (blue dashed arrows) in compound **3**.

**Table 1 molecules-24-00197-t001:** *K* values of target compounds in different solvent systems.

Solvent Systems	Ratio (*v*/*v*)	*K*				
		1	2	3	4	5
ethyl acetate/*n*-butanol/water	2:1:3	0.092	0.102	0.210	0.320	0.535
ethyl acetate/*n*-butanol/water	4:1:5	0.154	0.161	0.287	0.351	0.710
ethyl acetate/*n*-butanol/water	11:1:12	0.122	0.123	0.225	0.397	0.641
ethyl acetate/*n*-butanol/methanol/water	11:0.5:0.5:11	0.111	0.368	0.491	0.641	0.697
ethyl acetate/*n*-butanol/methanol/water	11:0.5:1:11	0.173	0.419	0.536	0.875	1.355
ethyl acetate/*n*-butanol/methanol/water	11:3:0.5:11	0.677	0.859	1.085	1.910	3.352
ethyl acetate/*n*-butanol/methanol/water	5:1:1:5	0.510	0.719	1.090	1.581	2.210

**Table 2 molecules-24-00197-t002:** ^1^H (600 MHz) NMR data of compound **1***^a^* and compounds **2**–**5**
*^b^* (*δ* in ppm, *J* in Hz).

No	1	2	3	4	5
1	5.40 s	5.50 d (5.4)	5.51 d (8.2)	5.50 d (5.2)	5.50 d (8.2)
3	7.63 d (1.2)	7.42 d (1.2)	7.57 s	7.42 d (0.9)	7.56 s
4	-	-	-	-	-
5	3.30 m	2.91 m	3.38 m	2.93 m	3.37 m
6	1.72 m, 1.82 m	1.63 ddd (12.8, 8.0, 4.5), 2.06 m	5.54 m	1.63 m, 2.07 m	5.54 m
7	5.41 s	4.49 dd (7.1, 4.5)	5.55 m	4.54 dd (6.8, 4.8)	5.50 m
8	5.48 ddd (17.3, 10.5, 9.9)	5.73 ddd (17.4, 10.2, 9.0)	5.72 ddd (17.3, 10.3, 8.5)	5.72 ddd (17.2, 10.3, 9.0)	5.71 ddd (17.3, 10.5, 8.6)
9	2.63 m	2.67 m	2.59 m	2.67 m	2.58 m
10	5.27 m, 5.29 m	5.26 brd (10.2), 5.30 brd (17.4)	5.19 brd (10.6), 5.22 brd (17.3)	5.26 brd (10.2), 5.30 brd (17.2)	5.18 brd (10.5), 5.22 brd (17.3)
11	-	-	-	-	-
1′	4.74 d (7.9)	4.67 d (7.9)	4.73 d (7.8)	4.67 d (7.9)	4.72 d (7.7)
2′	3.45 m	3.19 dd (8.7, 7.9)	3.20 dd (8.8, 7.8)	3.18 dd (8.9, 7.9)	3.19 dd (8.8, 7.6)
3′	3.60 t (8.5)	3.10–3.40 m	3.10–3.45 m	3.10–3.40 m	3.10–3.45 m
4′	3.41 t (8.5)	3.10–3.40 m	3.10–3.45 m	3.10–3.40 m	3.10–3.45 m
5′	3.65 m	3.10–3.40 m	3.10–3.45 m	3.10–3.40 m	3.10–3.45 m
6′	3.66 m, 3.97 m	3.66 dd (12.0, 6.0), 3.89 dd (12.0, 2.1)	3.67 m, 3.89 dd (11.8, 1.7)	3.65 dd (11.9, 6.0), 3.89 dd (11.9, 2.0)	3.66 m, 3.88 dd (11.9, 1.8)
11-OMe	-	3.69 s	3.68 s	3.69 s	3.67 s
1′′	3.92 m, 3.87 m	-	-	3.60 m, 3.40 m	-
2′′	-	-	3.06 m, 3.05 m	1.54 m	3.06 m
3′′	-	-	4.13 m, 4.11 m	1.41 m	4.07 m
4′′	-	-	1.24 t (7.4)	0.94 t (7.4)	1.61 m
5′′	-	-	-	-	1.39 m
6′′	-	-	-	-	0.95 t (7.4)
7-OMe_a_	-	3.28 s	-	3.29 s	-
7-OMe_b_	-	3.29 s	-	-	-

*^a^* NMR data were obtained in chloroform-*d*. *^b^* Data were measured in methanol-*d*_4_.

**Table 3 molecules-24-00197-t003:** ^13^C (150 MHz) NMR data of compound **1***^a^* and compounds **2**–**5 *^b^*** (*δ* in ppm).

No	1	2	3	4	5	No	1	2	3	4	5
1	97.1	97. 8	97. 4	97. 9	97.4	4′	70.2	71. 5	71. 5	71. 6	71.5
3	152.3	153. 2	154.2	153. 3	154.2	5′	76.1	78. 4	78. 5	78. 4	78.5
4	104.7	111.7	109.5	111.7	109.5	6′	65.3	62.8	62.8	62.8	62.8
5	22.0	29. 4	39.6	29. 3	39.6	11-OMe	-	51.7	51.8	51.7	51.7
6	29.5	33. 2	133. 5	33. 1	133.4	1′′	62.3	-	173.5	66.3	173.6
7	100.1	104.4	126.6	104.0	126.6	2′′	15.2	-	38.6	33.7	38.6
8	131.6	135. 8	135. 8	135. 8	135.8	3′′	-	-	61.8	20.5	65.6
9	42.4	45.3	46.3	45.4	46.3	4′′	-	-	14.5	14.3	31.8
10	121.2	119. 8	118. 8	119. 8	118.8	5′′	-	-	-	-	20.2
11	169.1	169. 1	168. 8	169. 1	168.7	6′′	-	-	-	-	14.1
1′	98.8	100.1	100.2	100.1	100.2	7-OMe_a_	-	53.9	-	53.8	-
2′	73.5	74. 6	74. 7	74. 7	74.8	7-OMe_b_	-	52.5	-	-	-
3′	75.8	78. 0	78. 0	78. 1	78.0						

*^a^* NMR data were obtained in chloroform-*d*. *^b^* Data were measured in methanol-*d*_4_.

**Table 4 molecules-24-00197-t004:** Anti-inflammatory activities of compounds **1**–**5**.

Compound	IC_50_ (µM)
**1**	5.90 ± 0.71
**2**	6.50 ± 1.10
**3**	4.52 ± 0.55
**4**	6.11 ± 0.93
**5**	5.35 ± 0.51
Ginkgolide B	2.21 ± 0.40

**Table 5 molecules-24-00197-t005:** Antibacterial activities of compounds **1**–**5** against *Staphylococcus aureus ATCC 25923*.

Compound	MIC (µg/mL)
**1**	17.5 ± 3.1
**2**	23.4 ± 4.0
**3**	15.4 ± 2.1
**4**	26.0 ± 3.7
**5**	13.7 ± 1.9
Penicillin	0.4 ± 0.1

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
