# Peer review of "Separation of Five Iridoid Glycosides from Lonicerae Japonicae Flos Using High-Speed Counter-Current Chromatography and Their Anti-Inflammatory and Antibacterial Activities"

_molecules, 2019, doi:10.3390/molecules24010197_

Round 1

Reviewer 1 Report

I have carefully read manuscript molecules-416071 entitled „Separation of five iridoid glycosides from Lonicerae Japonicae Flos using high-speed counter-current chromatography and their anti-inflammatory and antibacterial activities“. The scope of the paper is of interest and I have found a general good quality of the research. From experimentation to data evaluation, everything is well organized and clearly described. Chromatography separation, HRMS and NMR analysis appears to be carefully performed. In my opinion, this work could be accepted to be published in Molecules in present form.

Author Response

Dear reviewer,

      Thank you for your good comments on our manuscript(molecules-416071).

With Best Regards!

Yours sincerely,

Yongqing Zhang

Tel.: + 0531 89628803;

E-mail address: [email protected].

Reviewer 2 Report

The article describes separation and identification of 5 new compounds isolated from Lonicerae Japonicae flos. The HSCCC separation was optimised by selection of  K-value, for identification  the authors HR ESI MS and 2D NMR.

All the data presented form a logical structure, so that formally I can not object too much. 

Therefore only a few comments:

lines 63-71 (optimisation of HSCCC): the English might be improved - l.64 and 69 - is there a correct tense?, "big" seems not to be an appropriate expression

for readers who are nor familiar with HSCCC: the meaning of 48.5% of stationary phase retention should be perhaps explained by a few words 

Figure 2 (HSCCC chromatogram). While peaks I - V were identified; however no description of two peaks 200 and 250 min is given. Why? A brief explanation should be added

l. 93, 103 etc. data for specific rotation shoud contain an equal sign (CH3OH, c=1.0) or comma if notation "(c, 1.0 in CH3OH) would be chosen

Fig. 5: the quality of the figure was low in my copy

Table 4, Table 5: what was the precision of IC50 and MIC measurements? It should be given; otherwise the number of decimal digits should be reduced.

REFERENCES: l.252, 258,272,276,...etc latin names in the article tiles should be in italics 

Author Response

Dear reviewer,

Thank you for your good comments on our manuscript (molecules-416071). 

The manuscript entitled “Separation of five iridoid glycosides from Lonicerae Japonicae Flos using high-speed counter-current chromatography and their anti-inflammatory and antibacterial activities” has been carefully revised according to the comments.

Detailed response were listed as follows:

Point 1: Lines 63-71 (optimisation of HSCCC): the English might be improved - l.64 and 69 - is there a correct tense? "big" seems not to be an appropriate expression.

Response 1: Some changes have been made in the lines 63-71. The tense in l.64 and 69 have been corrected and "large" seems to be more appropriate than "big" for describing the K value.

Point 2: For readers who are nor familiar with HSCCC: the meaning of 48.5% of stationary phase retention should be perhaps explained by a few words.

Response 2: The meaning of 48.5% of stationary phase retention has been briefly explained in the revised manuscript using the sentence"...the stationary phase was 48.5% in this system, indicating  that there was sufficient stationary phase to allow the sample to be distributed." 

Point 3: Figure 2 (HSCCC chromatogram). While peaks I - V were identified; however no description of two peaks 200 and 250 min is given. Why? A brief explanation should be added.

Response 3: A brief explanation for peaks 200 and 250min have been added in the revised manuscript using this sentence "However, peaks 200 and 250 min contained more than one compound by HPLC analysis".

Point 4: L. 93, 103 etc. data for specific rotation should contain an equal sign (CH3OH, c=1.0) or comma if notation "(c, 1.0 in CH3OH) would be chosen.

Response 4: Equal signs have been added in the data for specific rotation.

Point 5: Fig. 5: the quality of the figure was low in my copy.

Response 5: The Fig.5 has been re-uploaded with 1200dpi.

Point 6: Table 4, Table 5: what was the precision of IC50 and MIC measurements? It should be given; otherwise the number of decimal digits should be reduced.

Response 6: The precision of IC50 and MIC measurements have been added in the revised manuscript.

Point 7: REFERENCES: l.252, 258,272,276,...etc latin names in the article tiles should be in italics .

Response 7: In the references, all the latin names in the article tiles have been revised to italics.

Thank you for your reviewing the revised manuscript!

With Best Regards!

Yours sincerely,

Yongqing Zhang

Tel.: + 0531 89628803;

E-mail address: [email protected].

Reviewer 3 Report

This is the manuscript entitled " Separation of five iridoid glycosides from Lonicerae Japonicae Flos using high-speed counter-current chromatography and their anti-inflammatory and antibacterial activities ”.

In the study the Authors described the glycosides present in Lonicerae Japonicae Flos using high-speed counter-current chromatography and their anti-inflammatory and antibacterial activities. The presence of different compounds (3) in the Lonicerae Japonicae Flos was previously confirmed in the literature. The structure two compound (3: adinoside F and 5: adinoside G) was described for the first time. Moreover, the authors were identified this compounds by HR-ESI-MS, 1D and 2D NMR.

In my opinion it is interesting for readers. The author studied the five iridoid glucosides in Lonicerae Japonicae Flos and gave a precision analysis of the ingredients by   HR-ESI-MS, 1D and 2D NMR. The experiment was designed well. The data were meaningful and richness.

Please check punctuation Page 1, Line 31.

Author Response

Dear reviewer,

Thank you for your good comments on our manuscript (molecules-416071). 

The manuscript entitled “Separation of five iridoid glycosides from Lonicerae Japonicae Flos using high-speed counter-current chromatography and their anti-inflammatory and antibacterial activities” has been carefully revised according to the comments.

Detailed response were listed as follows:

Point 1: Please check punctuation Page 1, Line 31.

Response 1: The punctuation in page 1, line 31 has been corrected.

Thank you for your reviewing the revised manuscript!

With Best Regards!

Yours sincerely,

Yongqing Zhang

Tel.: + 0531 89628803;

E-mail address: [email protected].